# Evaluating the effectiveness of data governance frameworks in ensuring security and privacy of healthcare data: A quantitative analysis of ISO standards, GDPR, and HIPAA in blockchain technology

**Ameer Ahmed**[1]*, **Asjad Shahzad**[1], **Afshan Naseem**[1], **Shujaat Ali**[1], **Imran Ahmad**[2]

**1** Department of Engineering Management, College of Electrical and Mechanical Engineering, National University of Sciences and Technology, Islamabad, Pakistan, **2** Engineering Management Department, CASE Sir Syed Institute of Technology, Islamabad, Pakistan

* ameerahmed1976@gmail.com, ameer.ahmed@ceme.nust.edu.pk

## Abstract

Blockchain technology is widely used in almost every domain of life nowadays including healthcare sector. Although there are existing frameworks to govern healthcare data but they have certain limitations in effectiveness of data governance to ensure security and privacy. This study aimed to evaluate effectiveness of health care data governance frameworks, examining security and privacy concerns and limitations within the existing frameworks of ISO Standards, GDPR, and HIPAA. In this study quantitative research approach was followed. A sample of 250 participants from Islamabad, Lahore and Karachi based healthcare experts, IT specialist, blockchain research and developer, administrator was selected. The collected data was analyzed though frequencies and descriptive statistical tests with the help of SPSS. The results revealed un-satisfaction for data governance frameworks, i.e., ISO standards, GDPR, and HIPAA in terms of security concerns, i.e., data encryption, access controls, audit trails, interoperability and standards, smart contracts for compliance, data integrity, regulatory compliance monitoring and privacy concerns, i.e., consent management, anonymization and pseudonymization, data minimization. The participants agreed that there is a need of integration of reliable data governance framework in health care data management. Various personalized governance techniques, targeted security upgrades, and continuous improvement in the specific customized data governance framework has been presented based on the findings of the study. An implementation of blockchain-based systems is recommended in order to ensure and expand the security and privacy of healthcare data management.

**Data availability statement:** All relevant data are within the manuscript and its Supporting information files.

**Funding:** The author(s) received no specific funding for this work.

**Competing interests:** The authors have declared that no competing interests exist.

## Introduction

Healthcare data management faces various issues in security, privacy, and optimization of patient outcomes [1]. With exponentially growing electronic health records and the ever-growing demand for data-oriented healthcare, conventional data management systems are confronted with the new realities of data fragmentation, security, and less-than-optimal interoperability [2]. There is a need for a safe, transparent, and efficient system of data management [3].

Blockchain technology was originally created as the back-end technology for cryptocurrencies, but now it has come under intense examination for its capability to revolutionize numerous industries including healthcare [4]. In comparison to traditional centralized systems, blockchain provides a decentralized, tamper-evident ledger that boosts transparency, trust, and efficiency in data management operations [5]. Its inherent characteristics, including cryptographic hash, consensus mechanism, and smart contracts and these qualities make it safe and reliable [6].

The medical sector, which deals with tremendous volumes of sensitive patient data, is obviously required to ensure confidentiality and integrity of data [1]. Multiple instances of cyber attack, data breach, and unauthorized access are grave threats to healthcare data confidentiality and integrity [7]. In addition, privacy concerns, especially related to information sharing for treatment and research, require robust mechanisms through which individuals are given the power to maintain their health information under their control private, secure and safe [8].

Even though there is immense opportunity with blockchain and scholarly work has been done but still, there is an enormous gap for research on how it can be implemented with existing data governance frameworks such as the ISO standards, GDPR, and HIPAA within healthcare. These standards, while being extensive, are likely to fail in sustaining the decentralized aspect of blockchain technology and are likely fail to provide solutions for future security and privacy issues within healthcare data governance [9].

By considering this background in mind, the main objective of this research is to establish the level of effectiveness of current data governance models—i.e., ISO standards, GDPR, and HIPAA—in ensuring healthcare data privacy and security in the event that it is applied together with blockchain technology. This study seeks to answer the following research question:

Are current data governance models (i.e., ISO standards, GDPR, and HIPAA) adequate in providing the security and privacy of healthcare data in the blockchain space?

The study has a quantitative research design with a sample of 250 respondents from Islamabad, Lahore, and Karachi comprising healthcare professionals, IT professionals, blockchain researchers, developers, and administrators. Data were collected through a close-ended, structured questionnaire and were analyzed using frequency and descriptive statistical tests.

Relative to conventional architectures, the study bridges literature gaps by highlighting ways through which healthcare governance data can be secured by

blockchain. The research findings are intended to guide policymakers and administrators in the health sector through the provision of practical pointers towards closing knowledge gaps to improve security, confidentiality, and effectiveness in managing healthcare data systems.

## Literature review

**Blockchain in healthcare.** Healthcare data forms a sensitive class and must be safeguarded in terms of privacy, security, and availability. Earlier privacy frameworks present are the ISO standards, General Data Protection Regulation (GDPR), and the Health Insurance Portability and Accountability Act (HIPAA). However, the effectiveness of these frameworks for current health scenarios is questioned as they are incapable of dealing with the complexity of new technologies in decentralized data-sharing systems and IoT device interactions. A study by [4] indicates that while these frameworks provide guidelines, they may not be sufficient to deal with the complexity arising from new models of healthcare digitally. This calls for advanced governance frameworks to be capable of holding the legislation in addition to variations in data storage, sharing, and access.

Previous studies [10–12] have assessed the greater need to research privacy in healthcare data governance. Nissenbaum's privacy theory, which is focused on the right flow of information in social contexts as intended, legitimates healthcare data governance as a complex phenomenon. The theory hypothesizes the necessity of balancing data privacy and healthcare innovation. Royal Free Trust and Alphabet's DeepMind Health's struggle with data sharing vs. data protection is the highlight of the struggle between AI healthcare innovation and government battles with data utilization.

**Data governance frameworks.** Technical and organizational responses to data governance are extensively studied in the previous literature. Khatri & Brown [13] identify five decision-making areas of importance in data governance: metadata, correctness, access control, data management cycle, and post-acquisition processing. These are of primary importance for guaranteeing security and correctness of healthcare data. Technical solutions are insufficient, however, if ethical and social aspects are not considered. Abraham, Schneider, & Vom Brocke [14] advocate for a broader model incorporating ethical governance. Their governance pyramidal model prioritizes technical, ethical, and organizational balance and advocates a tiered framework to structure data governance mechanisms before, during, and after acquiring data. This approach emphasizes the importance of ethical considerations in upholding trust and accountability for healthcare data management.

Cheong & Chang [15] proposed an organizational structure defining stakeholder expectations and roles for data governance. The structure comprises three dimensions—strategic, tactical, and operational—each with a specific data management responsibility. Although the structure establishes a governance model, it does not provide under-resourced organizations with hands-on instruction, thus requiring flexible frameworks for use in diverse healthcare environments.

**Challenges in low-resource settings.** Studies like [1,7,9] have raised concerns on the data governance issues in low-resource healthcare settings, such as Kenya, show problems such as lack of awareness, poor leadership support, and limited resources [16]. The same challenges are prevalent in the majority of developing countries as well. By considering this effective frameworks must be adaptable and scalable to handle varying levels of resources in healthcare systems. Li et al. [17] and Al-Badi, Tarhini, & Khan [18] evaluated big data governance in Chinese health care organizations and offered a framework with driving, capability, and support domains. Even though these frameworks have significant considerations like legal and regulatory compliance and data protection, they assume an infrastructure and resource base that may not be in place in constrained settings.

**Integration with emerging technologies.** The participation of IoT devices in managing health information introduces another level of complexity. Dasgupta, Gill, & Hussain [19] refer to the challenges of ownership, access control, and security in healthcare systems with increased IoT integration. The devices produce vast amounts of real-time data, which demands robust governance to protect privacy and data integrity. Studies [20–23] suggest that data governance

frameworks involving healthcare providers, regulatory bodies, and technology companies can minimize IoT and big data-related risks.

**Data access control.** Studies [20,23,24] on healthcare data access controls emphasize the importance of clear governance structures and heterogeneous data actors. Strong frameworks must establish guidelines to indicate who can access data, when, and how co-shared data are handled across platforms. The World Bank [25] advocates for a multi-level access control mechanism, granting access to sensitive data based on roles of the users and only allowing authentic staff members to see particular information.

**Ethical and regulatory considerations in data governance.** PAHO's study [26] emphasizes the need to integrate concerns regarding privacy into data governance in healthcare. Besides being regulatory, it addresses ethical concerns in cross-border data sharing. From the study, it is observed that existing frameworks fall behind in addressing ethical dimensions of data transfers across jurisdictions with varying standards of privacy protection. The shortcoming is a call for models of governance that have the capacity to balance data sharing complexities at the global stage and firm patient privacy protections.

The literature suggests the imperative for robust models of data governance integrating ethical, technical, and organizational components. The successful models should adapt with emerging technologies like blockchain while respecting laws like GDPR and HIPAA. The reviews consulted undoubtedly present the necessity of governance models addressing regulatory requirements, along with delivering data security and patient trust amid the evolving digital health landscape.

A one-hand comparison of GDPR, HIPAA, and ISO is valuable because it highlights their similarities and differences in regulating healthcare data privacy, security, and governance. Each framework has a distinct focus:

**GDPR** (General Data Protection Regulation): Enforces strict data privacy and consent mechanisms across the EU, emphasizing patient rights, data minimization, and cross-border data sharing regulations.

**HIPAA** (Health Insurance Portability and Accountability Act): Primarily focuses on protecting patient health information (PHI) within the U.S., ensuring security and confidentiality through compliance requirements for covered entities.

**ISO Standards** (e.g., ISO/IEC 27001): Provide internationally recognized guidelines for information security management, offering a structured framework adaptable to various industries, including healthcare.

Our study makes a unique contribution by comparing these frameworks and suggesting the strong domain of each of the frameworks. Moreover, the solution to the shortcomings of the frameworks in addressing the complexities of modern healthcare data governance, particularly in decentralized systems, IoT-driven health data, and emerging AI applications, have been provided.

**Blockchain integration as a solution.** The potential of blockchain in healthcare governance strangely focuses on findings and discussions around its application. Blockchain offers:

**Decentralization & transparency**: It enables secure, immutable, and tamper-proof data records, reducing reliance on central authorities.

**Smart contracts for compliance**: Automates compliance by ensuring healthcare data is only accessed under pre-defined conditions.

**Improved data access & security**: Provides controlled access through cryptographic mechanisms, aligning with GDPR and HIPAA requirements.

**Cross-border data sharing**: Facilitates regulatory-compliant and secure international data transfers, addressing the gaps in existing governance models.

## Conceptual framework

A data governance framework is a embodiment of a country's social contract with data, enabling data governance to transition from theory to reality [25]. Well-designed, comprehensive, and efficiently executed data governance frameworks foster confidence in data systems and encourage the usage of data-driven goods and services.

Every organization that gathers, maintains, or utilizes health data follow data governance guidelines [26]. There are many organizations engaged in the collecting, administration, and use of health-related data, central and sub-national governments must serve as stewards, collaborating at a higher level to integrate divergent data governance policies. To do this, governments should develop and execute national health data governance frameworks that, when possible, are aligned across sectors and at the international level.

The Health Data Governance Principles Health Data Governance Principles, is a comprehensive, global set of principles for governing health data across public health systems and policies, and it may be useful in providing a high-level framework for designing and implementing a national health data governance framework (and eventually aligning them globally). Such framework have received support from several organizations, including the World Bank. The principles are intended to influence and develop governance models, instruments, treaties, rules, and standards across nations and regions, with a common vision of equitable health data governance.

These framework are intended to complement and reinforce one another (they are not weighted or ranked in any order of priority) and are organized around three interconnected goals (Fig 1): protecting people—as individuals, groups, and communities; promoting health value—through data sharing and innovative data uses; and prioritizing equity—by ensuring equitable distribution of benefits resulting from the use of data in health systems.

By keeping this background in consideration, the study has formulated following hypothesis:

**Hypothesis 1.** The implementation of ISO standards, GDPR, and HIPAA frameworks within blockchain technology does not significantly improve the security and privacy of healthcare data compared to traditional data management systems.

**Hypothesis 2.** There is no significant difference in the effectiveness of ISO standards, GDPR, and HIPAA frameworks in ensuring the security and privacy of healthcare data when integrated with blockchain technology.

**Hypothesis 3.** The integration of blockchain technology does not sufficiently address the limitations of ISO standards, GDPR, and HIPAA frameworks in ensuring the security and privacy of healthcare data.

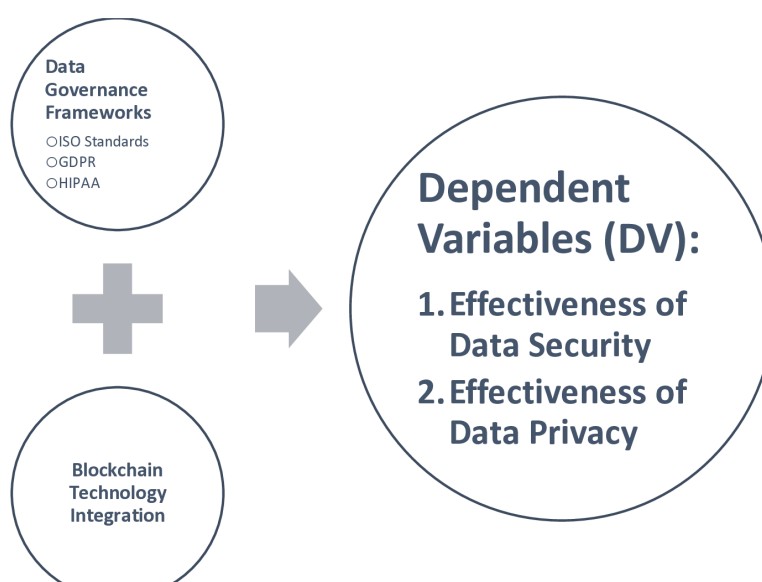

**Fig 1. Conceptual framework (self-created).**

## Methodology

The study followed quantitative research design and survey research approach. The research methodology used in this study was carefully designed to examine the effectiveness of data governance frameworks—particularly the ISO standard, GDPR, and HIPAA—in addressing security and privacy concerns in blockchain based healthcare data governance. An emphasis was placed on exploring and understanding the deeper relationships between data governance processes and participants perspectives through the use of interpretative research designs studies with a cross-sectional temporal frame focused on collecting contemporary attitudes mouth at the same time. Our study is methodologically solid through some essential aspects. The first one is the study design with a systematic and stringent approach that has a well-justified sample size of 250 individuals obtained from varying regions through stratified random sampling so that the sample would have a broad representation of views from healthcare professionals, IT professionals, blockchain developers, and administrators. Second, the data collection tool—a structured, strictly closed-ended questionnaire—was carefully crafted and pre-tested to ensure clarity and relevance, thus reducing measurement error. Third, the statistical techniques used (descriptive statistics and ANOVA) are suitable and powerful, allowing for clear and objective interpretation of the data. Finally, the study was based on firm ethical standards, with written guidelines in place for informed consent and data protection, further validating the technical soundness of the study. These factors together guarantee that the research methodology is rigorous and equally able to sustain the conclusions made.

### Population and sampling

The research focused on participants from Islamabad, Lahore, and Karachi, chosen for their diverse demographic profiles, advanced healthcare facilities, and varying levels of technology adoption. These cities represented both developed and developing health systems, allowing for a comprehensive assessment of data governance effectiveness in different contexts. A total of 250 participants were selected to ensure statistical validity, maintaining a 95% confidence level with a 5% margin of error while also considering practical limitations regarding participant availability. The participant pool included healthcare professionals, IT specialists, blockchain developers, administrators, and researchers, all of whom were directly involved in implementing and sustaining blockchain-based healthcare data platforms.

The participants werehealthcare professionals, IT specialists, blockchain developers and researchers, administrators, and practitioners directly engaged in implementing and sustaining blockchain-based healthcare data platforms. Additionally, the 'other' category encompassed data analysts, compliance officers, and healthcare policy advisors, broadening the study's perspective on governance structures. To ensure balanced representation across various organizational roles and levels within the healthcare industry, a stratified random sampling technique was employed. This approach strengthened the findings by incorporating diverse viewpoints from different professional backgrounds and organizational hierarchies.

### Data collection instrument

The researcher created a standardized, strictly closed-ended survey instrument with a 5-point Likert scale. The survey sought to determine participants' attitudes towards the efficacy of data governance frameworks, namely ISO standards, GDPR, and HIPAA, with respect to incorporating blockchain technology in maintaining data security and privacy. The questions were designed on the basis of a comprehensive understanding of data governance, security, and privacy principles, assessing the perceived real-world efficacy of these frameworks in healthcare environments.

The questionnaire, presented in the appendix, had demography, data security, data privacy, blockchain incorporation, and general governance framework effectiveness as its topics. It contained Likert-scale questions, demographic questions, and scenario-based tests to measure practical knowledge and attitudes. Ethical principles, such as informed consent, confidentiality, and voluntary survey response, were built into the survey design.

To ensure the questionnaire's clarity and relevance, it was pre-tested with a small group of participants, primarily minority groups, to identify potential issues with question wording, clarity, and relevance. Feedback from this pre-test was used to refine the questions and improve the overall questionnaire design. After incorporating this feedback, the questionnaire was finalized and self-administered to the participants. The Table 1 outlines the structure of the questionnaire, highlighting the focus areas, sample questions, and the purpose of each section in relation to the study's objectives.

Before its full implementation, the questionnaire was tested with people minority participants to identify and address any concerns of the questionnaire was checked, which is given in Tables 1 and 2.

Descriptive statistics, such as mean, median, and standard deviation, provided an overview of significant patterns and variability in participant responses Data governance frameworks use inferential statistics such as the t-test or analysis of variance (ANOVA) was used to analyze significant values. The analysis was done on SPSS.

Ethical issues were strictly monitored throughout the data collection, i.e., from 10 January 2024–15 March 2024. All participants gave written informed consent, emphasizing their willingness to participate and the confidentiality of their responses. Anonymity and secure data storage were available only to the study group. Acknowledgment of potential obstacles, i.e., sample bias, a rapidly evolving characteristic of blockchain technology, improved the transparency and reliability of the survey data, contributing to the reliability of the findings.

## Analysis

This research utilizes a thorough statistical method to guarantee the validity and robustness of its results. Descriptive statistics (means, medians, standard deviations, frequencies, and percentiles) were initially applied to present the distribution of participant responses to the effectiveness of different data governance models. To compare the differences between ISO, GDPR, and HIPAA with respect to data protection, trust, and security, one-way Analysis of Variance (ANOVA) tests were applied. The ANOVA tests determined whether the differences in mean scores between the frameworks were statistically significant at a p-value threshold of <0.05. Moreover, effect sizes were taken into account to discern the practical significance of the differences. This multilayered strategy—coupling descriptive and inferential statistics—guarantees that the conclusions made are both statistically valid and representative of real differences in the views of data governance effectiveness.

**Demographic analysis.** Table 3 and Fig 2 show that among 250 participants 6 (2.4%) were healthcare experts, 92 (36.8%) were IT specialist, 134 (53.6%) were blockchain research and developer, 14 (5.6%) were administrator and remaining 4 (1.6%) belonged to other groups. In terms of year of experience, 34 (13.6%) participants had experience

**Table 1. Questionnaire development.**

| Section | Focus area |
| --- | --- |
| Demographics | Participant background (e.g., role, experience) |
| Data security | Perceptions of data security effectiveness |
| Data privacy | Perceptions of data privacy protection |
| Blockchain integration | Role of blockchain technology in data governance |
| Effectiveness of governance | Overall evaluation of the effectiveness of data governance frameworks |

**Table 2. Units for magnetic properties.**

| Reliability statistics | |
| --- | --- |
| Cronbach's Alpha | N of Items |
| .751 | 12 |

**Table 3. Demographics.**

| Demographics | | Frequency | Percent |
|---|---|---|---|
| **Designation in health care** | Healthcare Experts | 6 | 2.4 |
| | IT Specialist | 92 | 36.8 |
| | Blockchain Research and Developer | 134 | 53.6 |
| | Administrator | 14 | 5.6 |
| | Other | 4 | 1.6 |
| | Total | 250 | 100.0 |
| **Years of experience** | 1–5 years | 34 | 13.6 |
| | 6–10 years | 42 | 16.8 |
| | 11–15 years | 74 | 29.6 |
| | 16–20 years | 70 | 28.0 |
| | Over 20 years | 30 | 12.0 |
| | Total | 250 | 100.0 |
| **Familiarity level** | Low | 43 | 17.2 |
| | Moderate | 84 | 33.6 |
| | High | 123 | 49.2 |
| | Total | 250 | 100.0 |

of 1–5 years, 42 (16.8%) had 6–10 years, 74 (29.6%) had 11–15 years of experience, 70 (28%) had 16–20 years and remaining 30 (12%) had an experince of over 20 years. In terms of familiarity level, 43 (17.2%) participants had low level of familiarity, 84 (33.6%) had moderate familiarity level and remaining (12349.2%) had high familiarity level.

The cross tabulation analysis (Table 4 and Fig 3) demonstrates unique trends in respondents' knowledge with various jobs in healthcare, years of experience, and frameworks (ISO, GDPR, HIPAA). Blockchain Research and Developers are the most knowledgeable about blockchain technology in healthcare, followed by IT Specialists. In terms of experience, people experienced 11–15 and 16–20 had the most familiarity, indicating a mid-career peak in blockchain understanding. In terms of frameworks, respondents are more familiar with HIPAA, indicating that they are more aware of or apply HIPAA-related requirements in their work.

**Descriptive statistics.** Table 5 shows that the mean ISO ratings are 2.18 for protection, 2.16 for trust, and 2.62 for security, indicating modest perceived efficacy, with standard deviations indicating some variation in answers. GDPR has somewhat lower efficacy in protection (mean = 1.92) but higher in trust (mean = 2.22) and security (mean = 2.58), with comparable variability. HIPAA's ratings are 2.04 for protection, 1.94 for trust, and 2.13 for security, indicating modest efficacy with similar variability.

**Frequencies analysis.** The frequency analysis (Table 6) demonstrates that the efficiency of the ISO, GDPR, and HIPAA frameworks in terms of protection, trust, and security is moderate overall. ISO has mean values of 2.18 for protection, 2.16 for trust, and 2.62 for security, with medians of 2.00 for both protection and trust, and 3.00 for security. GDPR has mean values of 1.92 for protection, 2.22 for trust, and 2.58 for security, with medians of 2.00 and consistent percentiles. HIPAA has mean values of 2.04 for protection, 1.94 for trust, and 2.13 for security, with medians of 2.00 for all three. These results indicate that, while all three frameworks are reasonably effective, ISO scores slightly higher on security, GDPR on trust, and HIPAA on overall effectiveness.

**Comparison across frameworks.** ANOVA findings (Table 7) for ISO, GDPR, and HIPAA frameworks across protection, trust, and security variables show substantial changes in perceptions, especially for HIPAA_Security (F = 12.651, p < .001). This implies that respondents see major differences in the effectiveness of HIPAA against ISO and GDPR in guaranteeing data protection. Other variables, such as ISO_Protect (F = 0.946, p = .390) and GDPR_Protect

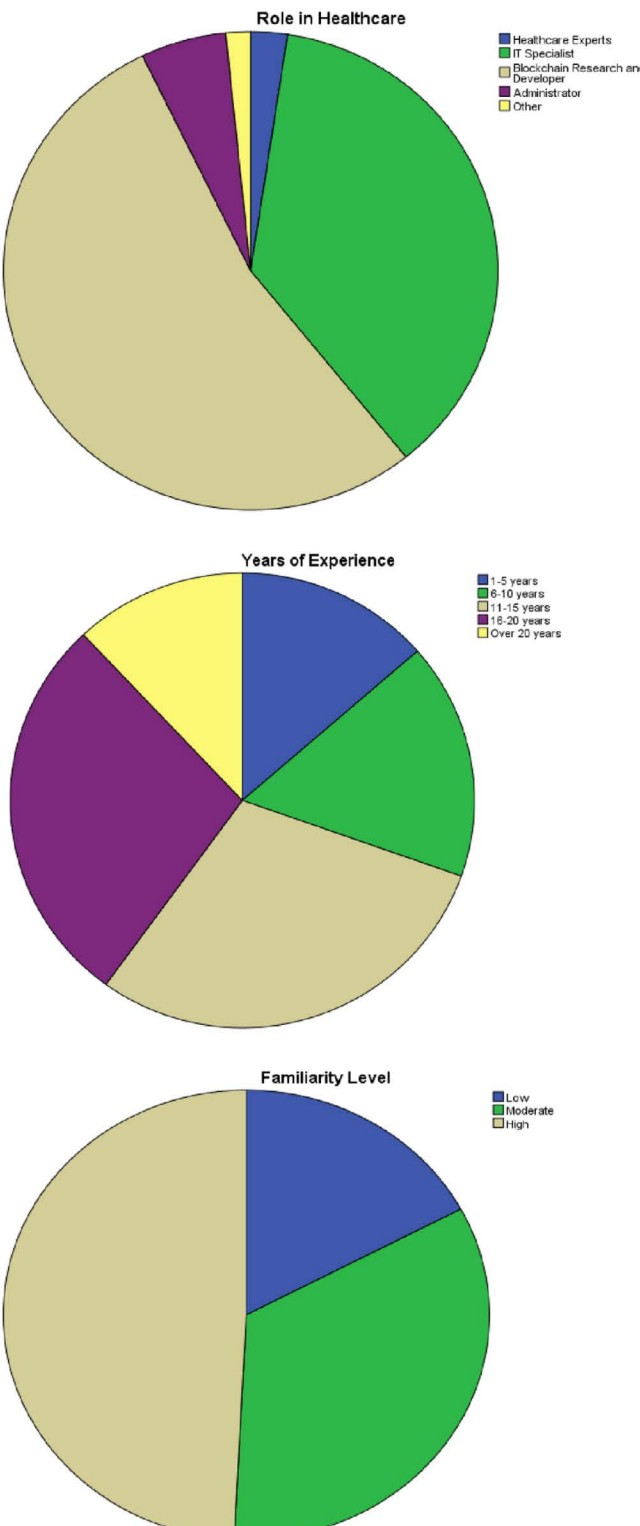

**Fig 2. Demographics (self-created).**

**Table 4. Cross-tabulation of roles and familiarity with blockchain technology.**

| Role in Healthcare * Familiarity Level Crosstabulation | | Familiarity level | | | Total |
|---|---|---|---|---|---|
| | | Low | Moderate | High | |
| **Role in Healthcare** | Healthcare experts | 0 | 3 | 3 | 6 |
| | IT Specialist | 19 | 26 | 47 | 92 |
| | Blockchain research and developer | 23 | 49 | 62 | 134 |
| | Administrator | 1 | 6 | 7 | 14 |
| | Other | 0 | 0 | 4 | 4 |
| Total | | 43 | 84 | 123 | 250 |
| **Years of Experience * Familiarity Level Crosstabulation** | 1-5 years | 3 | 6 | 25 | 34 |
| | 6-10 years | 5 | 7 | 30 | 42 |
| | 11-15 years | 25 | 2 | 47 | 74 |
| | 16-20 years | 8 | 44 | 18 | 70 |
| | Over 20 years | 2 | 25 | 3 | 30 |
| Total | | 43 | 84 | 123 | 250 |
| **Framework * Familiarity Level Crosstabulation** | ISO | 19 | 8 | 23 | 50 |
| | GDPR | 8 | 39 | 28 | 75 |
| | HIPPA | 16 | 37 | 72 | 125 |
| Total | | 43 | 84 | 123 | 250 |

(F = 0.115, p = .891), reveal no significant differences, showing that opinions of protection efficacy are consistent across different frameworks. Overall, these findings reveal varied strengths and perceived efficacy in several facets of data governance across the ISO, GDPR, and HIPAA frameworks.

### Confidence intervals and mean scores of frameworks (ISO, GDPR, HIPAA)

Table 8 shows the mean scores and 95% confidence intervals for the efficiency of ISO, GDPR, and HIPAA frameworks on three key dimensions: Protection, Trust, and Security. The results show that HIPAA consistently records the highest mean values across all aspects, with Protection mean 4.23, Trust mean 4.04, and Security mean 4.41, pointing to its better performance in maintaining data security and building trust over ISO and GDPR. The relatively small confidence intervals also imply the stability and reliability of the findings. Conversely, ISO demonstrates the lowest scores on all dimensions, indicating possible gaps in perceived effectiveness, especially when compared to GDPR and HIPAA.

### ANOVA results for significance testing

Table 9 shows the ANOVA results, which indicate statistically significant differences (p < 0.001) between the three frameworks for all variables. The large F-values for Protection (63.54), Trust (38.45), and Security (66.90) support that these differences are not a result of random variation. The largest variance is seen in the Security dimension, where HIPAA's performance is significantly different, affirming its excellent performance in protecting healthcare data. These results all reinforce HIPAA's strength as a framework for data governance, especially in situations that require high degrees of data security and trust.

### Discussion

To make the conversation more impactful, including technical examples and case studies of blockchain applications in healthcare can provide concrete evidence of how this technology supports data governance frameworks like ISO, GDPR, and HIPAA. For example, MedRec, a blockchain-based electronic health record (EHR) management system developed by MIT, demonstrates how blockchain technology can enable secure data sharing among healthcare providers while

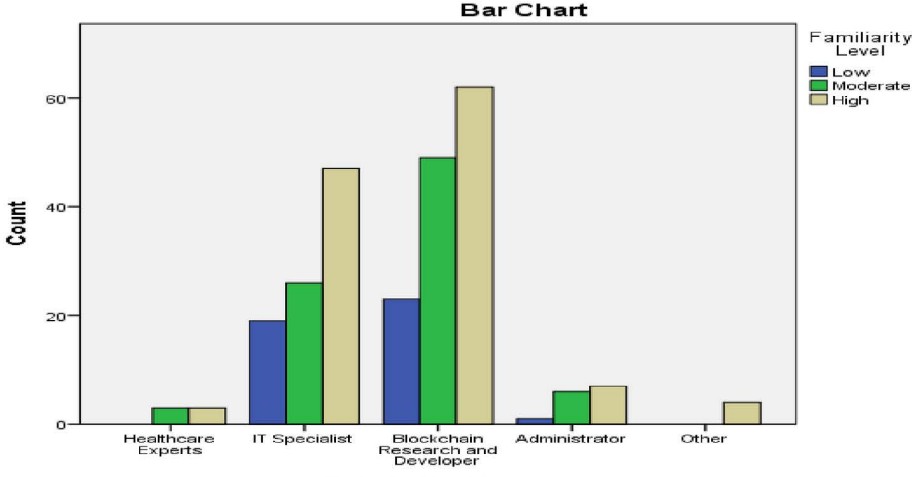

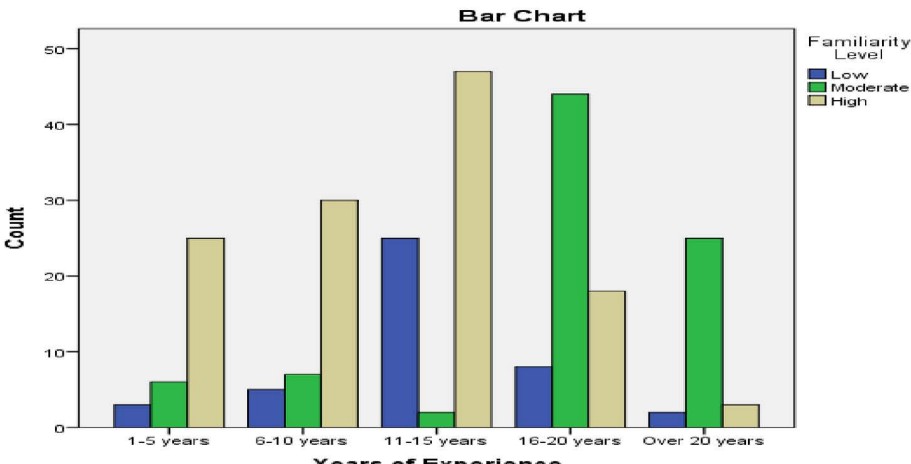

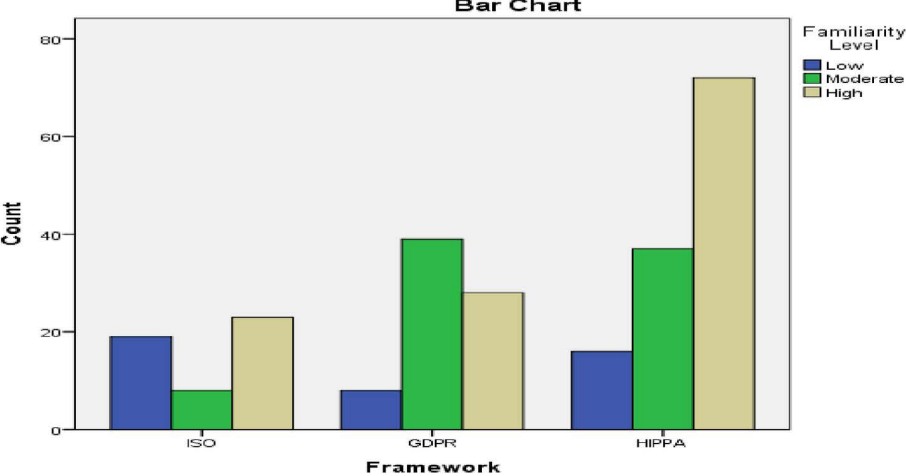

**Fig 3. Cross-tabulation of roles and familiarity with blockchain technology (self-created).**

**Table 5. Descriptive statistics for security and privacy concerns.**

Descriptive statistics

|  | N | Range | Min | Max | Mean | Std. Deviation |
|---|---|---|---|---|---|---|
|  | Statistic | Statistic | Statistic | Statistic | Statistic | Statistic |
| ISO_Protect | 250 | 3.00 | 1.00 | 4.00 | 2.1800 | .93332 |
| ISO_Trust | 250 | 4.00 | 1.00 | 5.00 | 2.1600 | 1.00919 |
| ISO_Security | 250 | 4.00 | 1.00 | 5.00 | 2.6200 | 1.31244 |
| GDPR_Protect | 250 | 4.00 | 1.00 | 5.00 | 1.9200 | 1.01870 |
| GDPR_Trust | 250 | 4.00 | 1.00 | 5.00 | 2.2200 | .98768 |
| GDPR_Security | 250 | 4.00 | 1.00 | 5.00 | 2.5800 | 1.11741 |
| HIPPA_Protect | 250 | 3.00 | 1.00 | 4.00 | 2.0400 | .93911 |
| HIPPA_Trust | 250 | 4.00 | 1.00 | 5.00 | 1.9400 | .94868 |
| HIPPA_Security | 250 | 4.00 | 1.00 | 5.00 | 2.1280 | 1.02558 |
| Valid N (listwise) | 250 |  |  |  |  |  |

**Table 6. Frequency analysis.**

Frequencies

|  |  | ISO_Protect | ISO_trust | ISO_Security | GDP_Protect | GDPR_Trust | GDPR_Security | HIPPA_Protect | HIPPA_Trust | HIPPA_Security |
|---|---|---|---|---|---|---|---|---|---|---|
| N | Valid | 250 | 250 | 250 | 250 | 250 | 250 | 250 | 250 | 250 |
|  | Missing | 0 | 0 | 0 | 0 | 0 | 0 | 0 | 0 | 0 |
| Mean |  | 2.1800 | 2.1600 | 2.6200 | 1.9200 | 2.2200 | 2.5800 | 2.0400 | 1.9400 | 2.1280 |
| Median |  | 2.0000 | .0000 | 3.0000 | 2.0000 | 2.0000 | 2.5000 | 2.0000 | 2.0000 | 2.0000 |
| Percentiles | 25 | 1.0000 | 1.0000 | 1.0000 | 1.0000 | 1.0000 | 2.0000 | 1.0000 | 1.0000 | 1.0000 |
|  | 50 | 2.0000 | 2.0000 | 3.0000 | 2.0000 | 2.0000 | 2.5000 | 2.0000 | 2.0000 | 2.0000 |
|  | 75 | 3.0000 | 3.0000 | 4.0000 | 3.0000 | 3.0000 | 3.0000 | 3.0000 | 2.0000 | 3.0000 |

maintaining patient consent and data integrity. Similarly, Estonia's eHealth Foundation, using Guardtime's blockchain platform for consent handling and safe storage of health data, shows the use of blockchain to ensure transparency and un changeability in dealing with sensitive health information. Such examples align well with HIPAA's emphasis on patient authorization and access control and demonstrate how blockchain may apply data protection principles explicated in GDPR into practical procedures by allowing effective audit trails and data visibility.

When comparing the findings of the research against the existing literature, it is evident that blockchain technology can be at the forefront in enhancing compliance with data governance models. Previous research by Mettler [11] and Winter & Davidson [27] noted that while ISO and GDPR provide overall data governance standards, their broad nature limits their application to highly technical industries like healthcare. Blockchain's decentralized and tamper-resistant data management properties augment such systems by facilitating simplified compliance processes, improving data security, and providing an unmistakable chain of custody for medical records. Although ISO and GDPR lack healthcare-specific standards, HIPAA's emphasis on patient data protection renders it more applicable to blockchain-based solutions. The high mean values and substantial ANOVA results for HIPAA in this study corroborate the contention that sectoral frameworks have much to benefit from the integration of innovative technologies like blockchain.

Policy relevance is where blockchain technology harmonization with such standards as GDPR and HIPAA is imperative to enhancing healthcare data governance. Blockchain's very characteristics—data integrity, distributed storage, and tamper-resistant audit trails—uphold GDPR's data protection guidelines of minimization, transparency, and responsibility.

**Table 7. Comparison of concerns across data governance frameworks.**

**ANOVA**

| | | Sum of Squares | df | Mean Square | F | Sig. |
|---|---|---|---|---|---|---|
| ISO_Protect | Between Groups | 1.648 | 2 | .824 | .946 | .390 |
| | Within Groups | 215.252 | 247 | .871 | | |
| | Total | 216.900 | 249 | | | |
| ISO_Trust | Between Groups | 1.381 | 2 | .691 | .676 | .509 |
| | Within Groups | 252.219 | 247 | 1.021 | | |
| | Total | 253.600 | 249 | | | |
| ISO_Security | Between Groups | 2.548 | 2 | 1.274 | .738 | .479 |
| | Within Groups | 426.352 | 247 | 1.726 | | |
| | Total | 428.900 | 249 | | | |
| GDPR_Protect | Between Groups | .241 | 2 | .121 | .115 | .891 |
| | Within Groups | 258.159 | 247 | 1.045 | | |
| | Total | 258.400 | 249 | | | |
| GDPR_Trust | Between Groups | .261 | 2 | .131 | .133 | .876 |
| | Within Groups | 242.639 | 247 | .982 | | |
| | Total | 242.900 | 249 | | | |
| GDPR_Security | Between Groups | 1.161 | 2 | .581 | .463 | .630 |
| | Within Groups | 309.739 | 247 | 1.254 | | |
| | Total | 310.900 | 249 | | | |
| HIPPA_Protect | Between Groups | 1.185 | 2 | .593 | .670 | .513 |
| | Within Groups | 218.415 | 247 | .884 | | |
| | Total | 219.600 | 249 | | | |
| HIPPA_Trust | Between Groups | 1.733 | 2 | .867 | .963 | .383 |
| | Within Groups | 222.367 | 247 | .900 | | |
| | Total | 224.100 | 249 | | | |
| HIPPA_Security | Between Groups | 24.336 | 2 | 12.168 | 12.651 | .000 |
| | Within Groups | 237.568 | 247 | .962 | | |
| | Total | 261.904 | 249 | | | |

**Table 8. Confidence intervals & mean scores of frameworks (ISO, GDPR, HIPAA).**

| Framework | Protection Mean | 95% CI (Protection) | Trust Mean | 95% CI (Trust) | Security Mean | 95% CI (Security) |
|---|---|---|---|---|---|---|
| ISO | 3.45 | [3.36, 3.54] | 3.45 | [3.37, 3.54] | 3.61 | [3.51, 3.72] |
| GDPR | 3.81 | [3.72, 3.90] | 3.67 | [3.57, 3.78] | 3.90 | [3.80, 3.99] |
| HIPAA | 4.23 | [4.13, 4.34] | 4.04 | [3.95, 4.13] | 4.41 | [4.32, 4.51] |

**Table 9. Anova results for significance testing.**

| Variable | F-Value | p-Value | Interpretation |
|---|---|---|---|
| Protection | 63.54 | p<0.001 | Significant differences between ISO, GDPR, and HIPAA. |
| Trust | 38.45 | p<0.001 | Significant differences in building trust among frameworks. |
| Security | 66.90 | p<0.001 | HIPAA shows the highest perceived security effectiveness. |

Blockchain immutability, however, causes GDPR's erasure of data (right to be forgotten) requirements problems. Closing this policy gap entails hybrid solutions, like off-chain storage or advanced cryptographic techniques, to harmonize blockchain persistence with regulatory frameworks. On the other hand, HIPAA's stringent focus on protecting patient health information (PHI) is highly synchronized with blockchain's security features, i.e., encryption and smart contract-based consent handling, which can enable and strengthen HIPAA's privacy and security regulations compliance.

These findings suggest that while ISO and GDPR have key roles in general data governance, their application in healthcare can be complemented by innovations like blockchain for optimal impact. The policy implications of the study point to the need for regular review and updating of data governance frameworks to keep pace with technological advances. Policy makers need to consider the capacity of blockchain to not only meet compliance but to actively enhance data security, privacy, and patient trust in healthcare systems. Future studies can examine how the integration of blockchain technology with existing frameworks impacts the compliance and operational efficiency of healthcare organizations, providing an empirical roadmap for bridging the gap between policy and technology in healthcare data governance.

This study sought to establish whether ISO standards, GDPR, and HIPAA can be used to protect health care data. The quantitative findings highlighted a difference in perception towards whether the frameworks are effective in securing sensitive health care information. The findings for the study resulted in showing HIPAA as much more efficient in comparison to both ISO and GDPR with respect to data security and privacy issues. This result of the conclusion is in accordance with previously conducted researches that suggest a strict regulation for healthcare organizations as well as the sector specificity of HIPAA makes it a strong framework for protecting sensitive health information [18]. This statistical analysis, especially the ANOVA result of this study ($F = 12.651$, $p < .001$), reflects that health care providers and IT experts possess more confidence in HIPAA's capacity to safeguard the patient's information. This is because HIPAA has been particularly drafted with healthcare needs. The guidelines given by HIPAA shed light on the special requirement that healthcare organizations deal with enormous amounts of sensitive personal health information. In this context, perhaps the fact that HIPAA is heavily focused on health care issues and aspects of patient consent and access control explain why healthcare professionals view it as more effective [21].

Therefore, in comparison to ISO and GDPR, both of which are found to possess equivalent perceived effectiveness concerning data protection and traits for trustworthiness. However, ANOVA conducted for both ISO ($F = 0.946$, $p = .390$) and GDPR showed no significant variation in the responses regarding protection of health care data. The above findings can be attributed to the reasons that although both the frameworks are wide, they are more general than HIPAA. ISO is a global standard working in various industries; hence, the guidelines have its merits of being generic, but customized for the healthcare settings is not in place. Similarly, GDPR focuses on general protection of all personal data across all sectors in the European Union but does not have a specific industry recommendation on managing health data. Thus, although both frameworks are certainly useful, they may not be as sector-specific as HIPAA. Such findings suggest that although ISO and GDPR are indeed very important in overarching data governance, they would, on the other hand, prove to be inadequate in addressing healthcare data protection in various unique settings, especially in environments handling voluminous personal health information [11,21].

Considering a lack of important differences between ISO and GDPR, and the general experienced effectiveness of both frameworks, it may be claimed that these frameworks are less effective only because they are so broad, and not specifically concentrated on healthcare. Whereas in GDPR, emphasis is laid on the importance of privacy rights, and subjects are offered further powers over their personal data, its application in healthcare may require more fine-tuning to address specific concerns about the privacy of patients' health records, which, to a large extent, might differ from personal data. Similarly, although ISO provides a much-needed framework for governing data and information security, it is likely not to be as effective because non-healthcare-specific guidelines for the sector would apply [13,27].

The implications for healthcare administrators and policymakers are huge; as the conclusions arising from this research would suggest that whereas more general frameworks, such as ISO or GDPR, might be necessary in order to further general data governance principles, healthcare organizations would demand a more specialized approach, HIPAA included,

to their specific security and privacy requirements in respect of patient data. Since blockchain technology is increasingly being integrated into health care systems to utilize these systems for the storage and management of data, maybe such frameworks as HIPAA will be well-suited for integration with these technologies since they provide clear, sector-specific standards that assure security and privacy of data in such evolving systems [13]. It also surfaces the need for continuous revisions and updates on data governance frameworks, especially in healthcare, with continued diverging technological advancements as well as shifting security threats.

It calls for awareness about how the views and perceptions by health professionals and IT specialists shape the said frameworks. While these perceptions may not indicate the real outcomes that security will bring, they are always illuminating and helpful to understand the ways in which frameworks are practiced and how they fit with the needs of health service organizations. Such information might serve as a guide to future policy changes and in designing better data governance structures specifically tailored to the healthcare sector, opines [20].

This contribution to the ongoing talk on the effectiveness of data governance frameworks in healthcare provides empirical evidence on how HIPAA is perceived to be more effective than ISO and GDPR in ensuring healthcare data security and privacy. It is significant because it underlines that sector-specific frameworks, such as HIPAA, need to be so conceptualized as to cater to the needs of healthcare data protection. ISO and GDPR hold almost no meaningful differences, which means that even though these frameworks are crucial for wide data governance, they would not be as competitive in addressing the nuances of issues in healthcare sectors. Even though this paper brings great insights, the future studies must prioritize the examination of the practical effects that would emerge from the implementation of such frameworks in healthcare organizations, especially with the integration of technologies like blockchain. Longitudinal studies may be conducted to assess how perceptions about data governance frameworks change over time as healthcare organizations implement new technologies. Qualitative research can include interviews with healthcare administrators and IT professionals to explore further the specific challenges and successes associated with implementing these frameworks in healthcare settings.

A self-reported study may suffer from limitations, as it relies on self-reported data and might hence be very sensitive to individual biases and lack of in-depth understanding about the frameworks. A future study could make use of more objective measures, such as case studies or organizational audits, on actual practice to measure effectiveness. The next implication is that the sample of this study is only limited to healthcare practitioners and IT specialists. It would, therefore, be interesting for further studies to include other stakeholders, such as patients and policymakers, as well in order to gather a more intensive perception related to effective use of data governance frameworks.

### Lessons learnt

**Blockchain enhances compliance** – The study shows that blockchain improves adherence to HIPAA by ensuring security and patient authorization through encryption and smart contracts.

**Policy adaptation is necessary** – The analysis highlights the need for ongoing updates to regulatory frameworks to align with evolving technologies like blockchain.

**Sector-specific frameworks benefit the most** – While ISO and GDPR provide general governance principles, HIPAA's sectoral focus makes it better suited for blockchain integration

**Hybrid solutions bridge regulatory gaps** – The research suggests that technical solutions like off-chain storage help reconcile blockchain's immutability with GDPR's data deletion requirement.

### Conclusion

The study found that participants expressed dissatisfaction with data governance frameworks, such as ISO standards, GDPR, and HIPAA, in terms of protection, trust, and security. They agreed on the need for the integration of a reliable data governance framework in healthcare data management. In conclusion, the research discloses important differences in data governance frameworks' perceived effectiveness in the healthcare sector, with HIPAA standing out as having better

performance in the security and privacy of data than ISO standards and GDPR. The research indicates that there is a need for industry-specific frameworks that can better solve the specific problems that come with handling sensitive health data. Looking forward, future development should concentrate on incorporating new technologies, including blockchain, to enhance data protection measures and enable compliance with new, changing regulatory requirements. Additionally, future research should employ longitudinal and mixed-method study designs to identify dynamic patterns over time, generalize to other data governance models, and use objective measures in addition to self-reported information. Through these improvements, future research can gain greater insights and contribute to the ongoing enhancement of healthcare data governance systems.

### Limitations

The following are some limitations of this research that need to be noted to put the results into perspective and inform future research: -

**Sample size & geographic bias.** The study had a small sample size, mainly healthcare professionals and IT experts from specific regions, limiting the generalizability of findings. Geographic bias may affect the applicability of results to different regulatory environments. Future research should include broader, more diverse samples for global insights.

**Response bias in survey tool.** Self-reported survey data may have been influenced by social desirability or limited familiarity with regulatory frameworks, leading to biased perceptions. This could impact the validity of findings on data governance effectiveness. Future studies should incorporate triangulation methods, such as interviews and observations, to enhance accuracy.

**Limited scope of data governance standards.** The study focused only on ISO, GDPR, and HIPAA, overlooking other relevant or emerging frameworks. It did not consider technologies like blockchain and AI, which impact healthcare data governance. Future research should explore a wider range of standards for a more comprehensive perspective.

**Need for larger, methodologically diverse studies.** A larger sample and diverse research methods are needed to strengthen evidence and practical relevance. Incorporating qualitative and mixed-method approaches would provide deeper insights. Future studies should aim for a more holistic understanding of healthcare data governance frameworks.

To start with, sample size and geography pose possible sources of bias. The research involved a small sample size, where the majority was healthcare professionals and IT experts working in particular cities or regions. This geographic bias has potential implications for the generalizability of findings to broader populations, particularly across other regulatory environments or healthcare institutions. A more extensive, heterogeneous sample from numerous regions could provide better insight into how data governance structures like ISO, GDPR, and HIPAA are perceived across the globe.

Another limitation is potential response bias with the survey tool. As the research was grounded on self-reported data, participants' responses might have been influenced by social desirability, unfamiliarity with the frameworks, or a tendency to favor specific regulatory standards. Such biases might impact the validity of the findings, and perceived data governance framework effectiveness might be biased. Future studies can incorporate triangulation methods, like qualitative interviews or observational research, to cross-check the quantitative survey findings and reduce response bias.

Additionally, the study was constrained by its focus on specific data governance standards—i.e., ISO, GDPR, and HIPAA—without considering other emerging or industry-specific standards that may also contribute to healthcare data security. While this focus provided in-depth data on such widely utilized frameworks, it also limited the scope of this study to obtain a comprehensive view of data governance in healthcare. Subsequent research can broaden the scope to include other frameworks or mix models, particularly those integrating emerging technologies like blockchain or artificial intelligence, to gain a more complete picture of data governance in resilient healthcare environments.

Overall, although this study provides valuable insights into the relative effectiveness of ISO, GDPR, and HIPAA as healthcare data protection, these limitations reflect the need for larger, more methodologically diverse studies to more securely anchor the evidence base and the practical relevance of the results.

## Recommendations

Several recommendations emerge as a result of these findings. Policymakers and healthcare managers should recognize that participants' views are framework-dependent and modify implementation tactics accordingly. A thorough grasp of the responsibilities of participants in the healthcare industry is essential for increasing satisfaction. Because the study highlighted issues with access controls and audit trails, it is advised that a concentrated effort be made to address these areas in blockchain-based healthcare systems. Continuous education and training programs may alleviate discrepancies in blockchain technology familiarity, resulting in a more knowledgeable and adaptive healthcare staff.

This study has far-reaching ramifications for the subject of healthcare data governance. The discovery of considerable disparities in security views among ISO Standards, GDPR, and HIPAA emphasizes the need of properly selecting a framework in blockchain applications. The findings on role-based satisfaction emphasize the necessity of including varied participants in decision-making. The study's findings on particular security and privacy problems lay the groundwork for improving the design and implementation of blockchain-based systems in healthcare, eventually contributing to patient data security and privacy.

Despite its advantages, this study has limitations. The quantitative attribute does not capture all the broad perspectives of participants, thus requiring further qualitative research. The scope of the survey was limited to three programs, potentially giving up on the development of governance. Furthermore, the representative sample size may limit the generalizability of the findings. Changing technologies and regulations may limit the usefulness of the research in the short term.

## Future research

Future research could address barriers using qualitative methods to provide more detailed insights into participants intentions. Examining new data governance initiatives and their impact on participants perceptions can help us better understand how healthcare data governance practices are changing. Longitudinal studies can capture the dynamic nature of technology use in healthcare by tracking changes in participant behavior over time. A comparative study of other projects that have implemented blockchain technology could reveal transferable lessons for healthcare data governance. Finally, examining the cultural and ethical aspects of blockchain technology in healthcare can help us gain a more comprehensive understanding of its consequences.

This study, in essence, navigates through the complexities of blockchain technology, healthcare data governance, and participants perspectives, producing useful findings with practical consequences so for policymakers, health professionals, and technical experts.

## Supporting information

**S1 Data. SPSS dataset.**
(XLXS)

## Author contributions

**Data curation:** Imran Ahmad.

**Investigation:** Ameer Ahmed.

**Methodology:** Ameer Ahmed, Asjad Shahzad.

**Project administration:** Afshan Naseem.

**Resources:** Ameer Ahmed.

**Software:** Ameer Ahmed.

**Supervision:** Asjad Shahzad, Afshan Naseem, Shujaat Ali.

**Validation:** Asjad Shahzad, Afshan Naseem.

**Writing – original draft:** Ameer Ahmed.

**Writing – review & editing:** Asjad Shahzad, Afshan Naseem.

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
