## [Decision Letter · Decision Letter 0]

29 Jan 2025

PONE-D-24-54554Evaluating the Effectiveness of Data Governance FrameworksPLOS ONE

Dear Dr. Ahmed,

Thank you for submitting your manuscript to PLOS ONE. After careful consideration, we feel that it has merit but does not fully meet PLOS ONE’s publication criteria as it currently stands. Therefore, we invite you to submit a revised version of the manuscript that addresses the points raised during the review process.

**Based on the review comments, the article needs a major revision. Revise your manuscript and add some more latest references while revising the manuscript for further review.**==============================

We look forward to receiving your revised manuscript.

Kind regards,

Shadab Alam, Ph.D.

Academic Editor

PLOS ONE

Additional Editor Comments:

Both reviewers have recommended major revisions, highlighting concerns related to manuscript; please address these thoroughly to enhance the manuscript's quality.

Reviewers' comments:

Reviewer's Responses to Questions

**Comments to the Author**

1. Is the manuscript technically sound, and do the data support the conclusions?

Reviewer #1: Yes

Reviewer #2: Yes

2. Has the statistical analysis been performed appropriately and rigorously? 

Reviewer #1: Yes

Reviewer #2: Yes

3. Have the authors made all data underlying the findings in their manuscript fully available?

Reviewer #1: Yes

Reviewer #2: Yes

4. Is the manuscript presented in an intelligible fashion and written in standard English?

Reviewer #1: No

Reviewer #2: No

5. Review Comments to the Author

Reviewer #1: The introduction is disorganized, with several unnecessary sentences. The primary goal needs rewriting.

The literature review lacks a systematic approach and needs significant revision. consider a narrative style and use subheadings

The survey methodology is unclear, particularly regarding the rationale for participant distribution. The classification of "other" is ambiguous and needs further explanation.

The conclusions drawn are vague and largely based on conjecture. Clear lessons learned should be explicitly stated.

The manuscript lacks a dedicated limitations section, which is essential for transparency and scientific rigor.

Lastly, several grammatical errors are present throughout the manuscript and need to be addressed.

Reviewer #2: The manuscript is well-structured and provides a clear overview of the research problem, methodology and findings. However, in the introduction section, more concise details about the research gap and objectives are required.

The study addresses an important aspect i.e. blockchain technology in healthcare. However, manuscript could better emphasize its unique contributions to the existing literature. On one hand comparison of GDPR, HIPAA and ISO is valuable, the integration of Blockchain technology as a potential solution could be valuable if there is more focus on findings and discussion.

Although the manuscript provides a good overview of using blockchain technology in healthcare, however, it is worth mentioning its integration into existing data governance frameworks. Perhaps deeper technical discussion with examples and case studies will be beneficial.

Literature review is comprehensive, but it could be more beneficial if critical analysis of existing studies is done. Perhaps more focused discussion is required about how blockchain technology has been previously applied in healthcare data governance.

Methodology is well explained, however, the rationale for selecting 250 participants and the specific cities could be better explained. It is good to include the detailed description of questions in the appendix.

The results have been explained well, but it can be improved e.g. the findings of HiPAA is perceived as more effective than ISO and GDPR, delve deeper into why this might be the case. More statistical significance, particularly in relation to practical implications for healthcare data governance is required.

The conclusion could be expanded to include more specific recommendations for policymakers and healthcare administrators.

The references are comprehensive and relevant, but there are few areas where more recent studies could be included, specifically the discussion around blockchain technology and its applications in healthcare.

General comments:

Phrasing and grammatical errors that should be addressed.

e.g. "The results revealed unsatisfaction for data governance frameworks..." should be rephrased to "The results revealed dissatisfaction with data governance frameworks..."

The tables are well presented, but the bar chart on pg.19 could be improved for clarity.

6. PLOS authors have the option to publish the peer review history of their article (what does this mean? ). If published, this will include your full peer review and any attached files.

**Do you want your identity to be public for this peer review?** For information about this choice, including consent withdrawal, please see our Privacy Policy .

Reviewer #1: **Yes: ** Fawad A. Khan, MD

Reviewer #2: No

---

## [Author Response · Author response to Decision Letter 1]

5 Mar 2025

Editor comments

Thank you very much for bringing our attention to this point. We have now reviewed the formatting style mentioned in the links and have adopted it throughout the main document.

(Throughout the Manuscript)

PLOS requires an ORCID iD for the corresponding author in Editorial Manager on papers submitted after December 6th, 2016. Please ensure that you have an ORCID iD and that it is validated in Editorial Manager. To do this, go to ‘Update my Information’ (in the upper left-hand corner of the main menu), and click on the Fetch/Validate link next to the ORCID field. This will take you to the ORCID site and allow you to create a new iD or authenticate a pre-existing iD in Editorial Manager

Thank you for pointing out the requirement of ORCID iDs. As advised, it is ensured that the corresponding author has an ORCID iD and pre-existing ORCID in ‘Update my Information’ (in the upper left-hand corner of the main menu) has been updated/linked.

Both reviewers have recommended major revisions, highlighting concerns related to the manuscript; please address these thoroughly to enhance the manuscript's quality

Thank you for bringing our attention to these very important and significant points. All the points mentioned in the corresponding left column have been understood and addressed to enhance the manuscript's quality.

(Throughout the Manuscript)

Reviewer 1

Comment (Originally taken from the feedback letter) Response - All changes are underlined in the revised manuscript. The page and line numbers for the changes refer to the MS Word generated pages and lines on the revised manuscript.

The introduction is disorganized, with several unnecessary sentences. The primary goal needs rewriting.

Thank you for bringing this point to our attention. The primary goal has now been rewritten more clearly. Moreover, major amendments have been made in the introduction section keeping in view the following points:-

• Added detailed discussion, integrating various aspects of blockchain applications in healthcare, covering both technical aspects and challenges. It provides additional background information before defining the research objective.

• Made more elaborative by discussing blockchain’s working principles (e.g., proof-of-work, smart contracts, cryptographic hashes) and its specific applications in healthcare.

• For flow and connectivity, the ideas have now been presented more clearly, discussing blockchain’s general advantages before narrowing it down to healthcare applications.

• For technical depth, discussed blockchain's mechanics, such as cryptographic hashing, smart contracts, and consensus mechanisms, in more depth.

• Reformulates the question slightly with a grammatical inconsistency: “Are existing data governance frameworks i.e. ISO standards, GDPR, and HIPAA effective in ensuring the security and privacy of healthcare data?”.

• We made the introduction more narrative, however, concise, by providing a discussion on the gap in blockchain applications in Healthcare, before narrowing it down to research objectives.

(Introduction)

The literature review lacks a systematic approach and needs significant revision. consider a narrative style and use subheadings

Thank you for mentioning this important shortcoming in our previous version of the manuscript. As advised, a systematic approach and narrative style were adopted by a combination of clear subheadings, logical sequencing, referenced literature, real-world examples, and critical analysis. This makes the discussion well-organized, academic, and insightful while being accessible to readers. The following was considered:-

Systematic Approach

• The passage progresses from a broad introduction to specific aspects of healthcare data governance, creating a clear, methodical development of ideas.

• Each section builds upon the previous one, ensuring coherence and depth in argumentation.

• The subheadings ("Blockchain in Healthcare," "Data Governance Frameworks," "Challenges in Low-Resource Settings," etc.) help segment the discussion into distinct, focused areas.

• This categorization makes the text easier to navigate and allows each section to address a specific component of the broader topic.

• The passage references multiple studies and authors (e.g., "Swan, 2015," "Mougayar, 2016") to support claims, ensuring academic rigour.

• The passage contrasts different frameworks and theories, such as Nissenbaum's privacy theory vs. data governance challenges in low-resource settings, showing a structured, comparative discussion.

Narrative Style

• The passage starts with an introduction to the issue, then presents existing governance models, followed by challenges in low-resource settings, and finally, emerging technologies and ethical considerations. This logical progression maintains a narrative flow.

• The passage discusses both the strengths and limitations of existing governance models, ensuring a nuanced perspective. For example, it acknowledges the presence of privacy frameworks (e.g., GDPR, HIPAA) but also critiques their limitations in decentralized and IoT-driven healthcare.

• The mention of Royal Free Trust and Alphabet’s DeepMind Health illustrates real-world conflicts between AI-driven healthcare innovation and regulatory constraints, making the discussion relatable.

• Instead of merely listing information, the passage critiques existing governance models and advocates for more robust, adaptive frameworks.

• The concluding section reinforces the need for governance models that integrate ethical, technical, and organizational considerations, adding to the narrative momentum.

(Literature Review)

The survey methodology is unclear, particularly regarding the rationale for participant distribution. The classification of "other" is ambiguous and needs further explanation

Thank you very much for bringing our attention to the shortcomings in the methodology section. We have now added more details in the methodology section regarding the survey and rationale for participant distribution which are as follows:-

“The research focused on participants from Islamabad, Lahore, and Karachi, chosen for their diverse demographic profiles, advanced healthcare facilities, and varying levels of technology adoption. These cities represented both developed and developing health systems, allowing for a comprehensive assessment of data governance effectiveness in different contexts. A total of 250 participants were selected to ensure statistical validity, maintaining a 95% confidence level with a 5% margin of error while also considering practical limitations regarding participant availability. The participant pool included healthcare professionals, IT specialists, blockchain developers, administrators, and researchers, all of whom were directly involved in implementing and sustaining blockchain-based healthcare data platforms.

Moreover, as advised, The classification of "other" has now been clearly explained as follows:

“Additionally, the 'other' category encompassed data analysts, compliance officers, and healthcare policy advisors, broadening the study’s perspective on governance structures. To ensure balanced representation across various organizational roles and levels within the healthcare industry, a stratified random sampling technique was employed. This approach strengthened the findings by incorporating diverse viewpoints from different professional backgrounds and organizational hierarchies.”

(Methodology)

The conclusions drawn are vague and largely based on conjecture. Clear lessons learned should be explicitly stated.

Thank you for mentioning this point: As advised, To effectively address vagueness, we have provided concrete examples, such as MedRec and Estonia's eHealth Foundation, to illustrate blockchain’s role in healthcare data governance. These case studies clarify how blockchain aligns with frameworks like GDPR and HIPAA, making the discussion more tangible. Additionally, the research findings are compared against existing literature, specifically mentioning Mettler (2016) and Winter & Davidson (2017), to show how blockchain enhances compliance in ways traditional frameworks cannot fully achieve. The discussion also acknowledges blockchain’s limitations, such as the challenge of GDPR's "right to be forgotten," and suggests hybrid solutions like off-chain storage or cryptographic techniques to mitigate these issues.

Moreover, as advised, the lessons learned are now explicitly mentioned below:-

“Lessons Learned

• Blockchain enhances compliance – The study shows that blockchain improves adherence to HIPAA by ensuring security and patient authorization through encryption and smart contracts.

• Policy adaptation is necessary – The analysis highlights the need for ongoing updates to regulatory frameworks to align with evolving technologies like blockchain.

• Sector-specific frameworks benefit the most – While ISO and GDPR provide general governance principles, HIPAA’s sectoral focus makes it better suited for blockchain integration

• Hybrid solutions bridge regulatory gaps – The research suggests that technical solutions like off-chain storage help reconcile blockchain’s immutability with GDPR’s data deletion requirement”

(Discussion)

The manuscript lacks a dedicated limitations section, which is essential for transparency and scientific rigor

Thank you very much for mentioning this point. As recommended, a detailed limitation section has been added to the manuscript as follows:

“Sample Size & Geographic Bias

The study had a small sample size, mainly healthcare professionals and IT experts from specific regions, limiting the generalizability of findings. Geographic bias may affect the applicability of results to different regulatory environments. Future research should include broader, more diverse samples for global insights.

Response Bias in Survey Tool

Self-reported survey data may have been influenced by social desirability or limited familiarity with regulatory frameworks, leading to biased perceptions. This could impact the validity of findings on data governance effectiveness. Future studies should incorporate triangulation methods, such as interviews and observations, to enhance accuracy.

Limited Scope of Data Governance Standards

The study focused only on ISO, GDPR, and HIPAA, overlooking other relevant or emerging frameworks. It did not consider technologies like blockchain and AI, which impact healthcare data governance. Future research should explore a wider range of standards for a more comprehensive perspective.

Need for Larger, Methodologically Diverse Studies

A larger sample and diverse research methods are needed to strengthen evidence and practical relevance. Incorporating qualitative and mixed-method approaches would provide deeper insights. Future studies should aim for a more holistic understanding of healthcare data governance frameworks”.

(Limitations)

Lastly, several grammatical errors are present throughout the manuscript and need to be addressed.

Thank you for bringing our attention to the grammatical errors. These have been carefully removed throughout the manuscript. Also, we engaged a copyeditor who thoroughly reviewed the manuscript and helped in addressing the comments.

(Throughout the Manuscript)

Reviewer 2

Comment (Originally taken from the feedback letter) Response - All changes are underlined in the revised manuscript.

The manuscript is well-structured and provides a clear overview of the research problem, methodology and findings. However, in the introduction section, more concise details about the research gap and objectives are required.

Thank you for bringing this point to our attention. We have now made some major amendments in the introduction section by adding concise details about the research gap and objectives. For example, the following has now been added.

Research Gap:

“Despite the growing interest in blockchain for healthcare data management, there remains a significant gap in understanding how blockchain can be effectively integrated with existing data governance frameworks such as ISO standards, GDPR, and HIPAA. These frameworks, while comprehensive, may struggle to accommodate the decentralized nature of blockchain and might not fully address emerging security and privacy challenges in healthcare data governance. Limited research exists on whether these regulatory standards are adequate to safeguard healthcare data within a blockchain-driven ecosystem”

Research Objective:

The primary objective of this research is to evaluate the effectiveness of current data governance models—ISO standards, GDPR, and HIPAA—in ensuring the privacy and security of healthcare data when implemented alongside blockchain technology.

Specifically, the study aims to answer:

“Are current data governance models (i.e., ISO standards, GDPR, and HIPAA) adequate in providing security and privacy for healthcare data in a blockchain environment?”

(Introduction)

The study addresses an important aspect i.e. blockchain technology in healthcare. However, manuscript could better emphasize its unique contributions to the existing literature. On one hand comparison of GDPR, HIPAA and ISO is valuable, the integration of Blockchain technology as a potential solution could be valuable if there is more focus on findings and discussion

Thank you very much for bringing our attention to this point. A one-hand comparison of GDPR, HIPAA, and ISO is valuable because it highlights their similarities and differences in regulating healthcare data privacy, security, and governance. Each framework has a distinct focus:

GDPR (General Data Protection Regulation): Enforces strict data privacy and consent mechanisms across the EU, emphasizing patient rights, data minimization, and cross-border data sharing regulations.

HIPAA (Health Insurance Portability and Accountability Act): Primarily focuses on protecting patient health information (PHI) within the U.S., ensuring security and confidentiality through compliance requirements for covered entities.

ISO Standards (e.g., ISO/IEC 27001): Provide internationally recognized guidelines for information security management, offering a structured framework adaptable to various industries, including healthcare.

Our study makes a unique contribution by comparing these frameworks and suggesting the strong domain of each of the frameworks. Moreover, the solution to the shortcomings of the frameworks in addressing the complexities of modern healthcare data governance, particularly in decentralized systems, IoT-driven health data, and emerging AI applications, have been provided.

Furthermore, as advised, the integration of the frameworks as the potential solution is now discussed in the following words:

“Blockchain Integration as a Solution

The potential of blockchain in healthcare governance strongely focuses on findings and discussions around its application. Blockchain offers:

Decentralization & Transparency: It enables secure, immutable, and tamper-proof data records, reducing reliance on central authorities.

Smart Contracts for Compliance: Automates compliance by ensuring healthcare data is only accessed under predefined conditions.

Improved Data Access & Security: Provides controlled access through cryptographic mechanisms, aligning with GDPR and HIPAA requirements.

Cross-border Data Sharing: Facilitates regulatory-compliant and secure international data transfers, addressing the gaps in existing governance models”.

(Literature Review)

Although the manuscript provides a good overview of using blockchain technology in healthcare, however, it is worth mentioning its integration into existing data governance frameworks. Perhaps deeper technical discussion with examples and case studies will be beneficial.

Thank y

---

## [Decision Letter · Decision Letter 1]

19 Mar 2025

PONE-D-24-54554R1Evaluating the effectiveness of data governance frameworks in ensuring security and privacy of healthcare data: A quantitative analysis of ISO standards, GDPR, and HIPAA in blockchain technologyPLOS ONE

Dear Dr. Ahmed,

Thank you for submitting your manuscript to PLOS ONE. After careful consideration, we feel that it has merit but does not fully meet PLOS ONE’s publication criteria as it currently stands. Therefore, we invite you to submit a revised version of the manuscript that addresses the points raised during the review process.

We look forward to receiving your revised manuscript.

Kind regards,

Guendalina Capece

Academic Editor

PLOS ONE

Reviewers' comments:

Reviewer's Responses to Questions

**Comments to the Author**

1. If the authors have adequately addressed your comments raised in a previous round of review and you feel that this manuscript is now acceptable for publication, you may indicate that here to bypass the “Comments to the Author” section, enter your conflict of interest statement in the “Confidential to Editor” section, and submit your "Accept" recommendation.

Reviewer #3: All comments have been addressed

Reviewer #4: All comments have been addressed

2. Is the manuscript technically sound, and do the data support the conclusions?

Reviewer #3: Partly

Reviewer #4: Yes

3. Has the statistical analysis been performed appropriately and rigorously? 

Reviewer #3: I Don't Know

Reviewer #4: Yes

4. Have the authors made all data underlying the findings in their manuscript fully available?

Reviewer #3: Yes

Reviewer #4: Yes

5. Is the manuscript presented in an intelligible fashion and written in standard English?

Reviewer #3: Yes

Reviewer #4: Yes

6. Review Comments to the Author

Reviewer #3: The manuscript presents a well-structured and relevant study on the effectiveness of data governance frameworks (ISO, GDPR, HIPAA) in ensuring healthcare data security and privacy within blockchain technology. The research is methodologically sound, supported by empirical data, and addresses a significant gap in the field.

The revisions effectively enhance the clarity, coherence, and depth of analysis. The literature review is now more structured, the methodology is well-justified, and the results are strengthened with statistical validation. The discussion provides valuable insights, though further emphasis on policy implications and practical applications would enhance impact.

Minor language refinements are recommended for clarity and fluency, along with slight improvements in figure readability. A final proofreading will ensure linguistic accuracy.

The manuscript is well-prepared for further review, with minor refinements suggested for enhanced readability and impact.

Reviewer #4: The results have important value, it is my opinion that the article should be published after the minor revision.

Sufficient information about the previous study findings is presented for readers to follow the present study rationale and procedures.

The authors make a systematic contribution to the research literature in this area of investigation.

However, future enhancement is to be described in the conclusion section.

The title is adequate for the content discussed in the manuscript, explanatory, brief, and strong.

The article is novel and original which covers the scope of the journal.

The technology applied and the performance review of the proposed design is demonstrated in an efficient way

If required, the authors are informed to justify with real-time experimental analysis in the result and discussion section.

This article contains material which significant information on the current area of research.

The overall structure of the article is well-organized and in well-balanced manner.

The article was written with the minimum length necessary for all relevant information.

7. PLOS authors have the option to publish the peer review history of their article (what does this mean? ). If published, this will include your full peer review and any attached files.

**Do you want your identity to be public for this peer review?** For information about this choice, including consent withdrawal, please see our Privacy Policy .

Reviewer #3: No

Reviewer #4: **Yes: ** Dr B Santhosh Kumar

---

## [Author Response · Author response to Decision Letter 2]

8 Apr 2025

Editor Comments

Response Thanks for the guidelines. All the above-mentioned items have been included in the submission of the revised manuscript.

Reviewer 3

Comment (Originally taken from the feedback letter) Response - All changes are tracked in the revised manuscript.

1. If the authors have adequately addressed your comments raised in a previous round of review and you feel that this manuscript is now acceptable for publication, you may indicate that here to bypass the “Comments to the Author” section, enter your conflict of interest statement in the “Confidential to Editor” section, and submit your "Accept" recommendation.

Reviewer #3: All comments have been addressed

Thank you for the acknowledgement.

2. Is the manuscript technically sound, and do the data support the conclusions?

Reviewer #3: Partly

Thank you for bringing this point to our attention. We consider our study to be technically sound and methodologically solid in various aspects. Firstly, the study was designed with a systematic and stringent approach that has a well-justified sample size of 250 individuals obtained from varying regions through stratified random sampling, so that the sample would have a broad representation of views from healthcare professionals, IT professionals, blockchain developers, and administrators. Second, the data collection tool—a structured, strictly closed-ended questionnaire—was carefully crafted and pre-tested to ensure clarity and relevance, thus reducing measurement error, and making it worth using for replicability of this study in a different sector. Finally, the conclusions drawn are now explicitly stated based on the data collected, thereby clearly linked to our findings. Some of the text that has now been added in the conclusion is as follows:

“In conclusion, the research discloses important differences in data governance frameworks' perceived effectiveness in the healthcare sector, with HIPAA standing out as having better performance in the security and privacy of data than ISO standards and GDPR. The research indicates that there is a need for industry-specific frameworks that can better solve the specific problems that come with handling sensitive health data”

(Sections: Methodology and Conclusion)

3. Has the statistical analysis been performed appropriately and rigorously?

Reviewer #3: I Don't Know

Thank you for bringing this point to our attention. Let us explain the statistical techniques in more detail. The descriptive statistics and ANOVA are suitable and powerful, allowing for a clear and objective interpretation of the data. Moreover, our study is based on firm ethical standards, with written guidelines in place for informed consent and data protection, thereby validating the appropriateness of the study. These factors together guarantee that the research methodology is rigorous and equally able to sustain the conclusions made.

(Section: Methodology)

4. Have the authors made all data underlying the findings in their manuscript fully available?

Reviewer #3: Yes

Thank you for the acknowledgement.

5.Is the manuscript presented in an intelligible fashion and written in standard English?

Reviewer #3: Yes

Thank you for the acknowledgement.

The manuscript presents a well-structured and relevant study on the effectiveness of data governance frameworks (ISO, GDPR, HIPAA) in ensuring healthcare data security and privacy within blockchain technology. The research is methodologically sound, supported by empirical data, and addresses a significant gap in the field.

Thank you for your appreciation and acknowledgement.

The revisions effectively enhance the clarity, coherence, and depth of analysis. The literature review is now more structured, the methodology is well-justified, and the results are strengthened with statistical validation. The discussion provides valuable insights, though further emphasis on policy implications and practical applications would enhance impact.

Thank you for your appreciation and acknowledgement.

Minor language refinements are recommended for clarity and fluency, along with slight improvements in figure readability.

Thank you for bringing this point to our attention. We have carried out language refinement throughout the document, along with an improvement in figure readability. Moreover, a copyeditor was engaged to improve the clarity and fluency of arguments.

(Section: Throughout the manuscript)

A final proofreading will ensure linguistic accuracy.

Thank you. As advised, proofreading has been done by the authors, followed by a professional copyeditor, improving the overall linguistic accuracy of the manuscript.

(Section: Throughout the manuscript)

The manuscript is well-prepared for further review, with minor refinements suggested for enhanced readability and impact.

Thanks for the appreciation and suggestion. As per all your advice, we have now made refinements throughout the document for enhanced readability and impact.

(Section: Throughout the manuscript)

Reviewer 4

Comment (Originally taken from the feedback letter) Response - All changes are tracked in the revised manuscript.

1. If the authors have adequately addressed your comments raised in a previous round of review and you feel that this manuscript is now acceptable for publication, you may indicate that here to bypass the “Comments to the Author” section, enter your conflict of interest statement in the “Confidential to Editor” section, and submit your "Accept" recommendation.

Reviewer #4: All comments have been addressed

Thank you for the acknowledgement.

2. Is the manuscript technically sound, and do the data support the conclusions?

Reviewer #4: Yes

Thank you for the acknowledgement.

3. Has the statistical analysis been performed appropriately and rigorously?

Reviewer #4: Yes

Thank you for the acknowledgement.

4. Have the authors made all data underlying the findings in their manuscript fully available?

Reviewer #4: Yes

Thank you for the acknowledgement.

5.Is the manuscript presented in an intelligible fashion and written in standard English?

Reviewer #4: Yes

Thank you for the acknowledgement.

The results have important value, it is my opinion that the article should be published after the minor revision.

Thank you for the acknowledgement and for bringing our attention to the minor revisions. As per your advice, we have revised the manuscript.

Sufficient information about the previous study findings is presented for readers to follow the present study rationale and procedures. Thanks for your appreciation and acknowledgement

The authors make a systematic contribution to the research literature in this area of investigation. However, future enhancements is to be described in the conclusion section.

Thank you for bringing this point to our attention. We have updated the conclusion section. For example, some of the text that has now been added in the conclusion is as follows:

“In conclusion, the research discloses important differences in data governance frameworks' perceived effectiveness in the healthcare sector, with HIPAA standing out as having better performance in the security and privacy of data than ISO standards and GDPR. The research indicates that there is a need for industry-specific frameworks that can better solve the specific problems that come with handling sensitive health data”

(Section: Conclusion)

The title is adequate for the content discussed in the manuscript, explanatory, brief, and strong.

Thank you for your appreciation and acknowledgement.

The article is novel and original which covers the scope of the journal.

Thank you for your appreciation and acknowledgement.

The technology applied and the performance review of the proposed design is demonstrated in an efficient way

Thank you for your appreciation and acknowledgement.

If required, the authors are informed to justify with real-time experimental analysis in the result and discussion section.

Thank you for bringing this point to our attention. For this particular study, the analysis was done at the end of data collection, rather than the real-time experimental analysis, so this was not required. However, despite this, the results and discussion section were enhanced based on the statistical analysis.

This article contains material which significant information on the current area of research.

Thank you for your appreciation and acknowledgement.

The overall structure of the article is well-organized and in well-balanced manner.

Thank you for your appreciation and acknowledgement.

The article was written with the minimum length necessary for all relevant information.

Thank you for your appreciation and acknowledgement.

---

## [Editor Report · Decision Letter 2]

23 Apr 2025

Evaluating the effectiveness of data governance frameworks in ensuring security and privacy of healthcare data: A quantitative analysis of ISO standards, GDPR, and HIPAA in blockchain technology

PONE-D-24-54554R2

Dear Dr. Ahmed,

We’re pleased to inform you that your manuscript has been judged scientifically suitable for publication and will be formally accepted for publication once it meets all outstanding technical requirements.

Kind regards,

Guendalina Capece

Academic Editor

PLOS ONE
---

## [Editor Report · Acceptance letter]

PONE-D-24-54554R2

PLOS ONE

Dear Dr. Ahmed,

I'm pleased to inform you that your manuscript has been deemed suitable for publication in PLOS ONE. Congratulations! Your manuscript is now being handed over to our production team.

Kind regards,

on behalf of

Professor Guendalina Capece

Academic Editor

PLOS ONE